# A Novel Predictor Tool of Biochemical Recurrence after Radical Prostatectomy Based on a Five-MicroRNA Tissue Signature

**DOI:** 10.3390/cancers11101603

**Published:** 2019-10-21

**Authors:** Zhongwei Zhao, Sabine Weickmann, Monika Jung, Michael Lein, Ergin Kilic, Carsten Stephan, Andreas Erbersdobler, Annika Fendler, Klaus Jung

**Affiliations:** 1Department of Urology, Charité-Universitätsmedizin Berlin, 10117 Berlin, Germany; zhaozhongweixy@163.com (Z.Z.); sabine.weickmann@charite.de (S.W.); mchjung94@gmail.com (M.J.); carsten.stephan@charite.de (C.S.); 2Department of Urology, Qilu Hospital of Shandong University, Jinan 250012, Shandong, China; 3Department of Urology, Sana Medical Center Offenbach, 63069 Offenbach am Main, Germany; michael.lein@sana.de; 4Berlin Institute for Urologic Research, 10115 Berlin, Germany; 5Institute of Pathology, Hospital Leverkusen, 51375 Leverkusen, Germany; e.kilic@pathologie-leverkusen.de; 6Institute of Pathology, University of Rostock, 18055 Rostock, Germany; andreas.erbersdobler@med.uni-rostock.de; 7Max Delbrueck Center for Molecular Medicine in the Helmholtz Association, Cancer Research Program, 13125 Berlin, Germany; annika.fendler@mdc-berlin.de

**Keywords:** microRNAs, prostate cancer, radical prostatectomy, tissue-based biomarkers, prognostic biomarkers, biochemical recurrence, prediction accuracy, predictive models

## Abstract

Within five to ten years after radical prostatectomy (RP), approximately 15–34% of prostate cancer (PCa) patients experience biochemical recurrence (BCR), which is defined as recurrence of serum levels of prostate-specific antigen >0.2 µg/L, indicating probable cancer recurrence. Models using clinicopathological variables for predicting this risk for patients lack accuracy. There is hope that new molecular biomarkers, like microRNAs (miRNAs), could be potential candidates to improve risk prediction. Therefore, we evaluated the BCR prognostic capability of 20 miRNAs, which were selected by a systematic literature review. MiRNA expressions were measured in formalin-fixed, paraffin-embedded (FFPE) tissue RP samples of 206 PCa patients by RT-qPCR. Univariate and multivariate Cox regression analyses were performed, to assess the independent prognostic potential of miRNAs. Internal validation was performed, using bootstrapping and the split-sample method. Five miRNAs (miR-30c-5p/31-5p/141-3p/148a-3p/miR-221-3p) were finally validated as independent prognostic biomarkers. Their prognostic ability and accuracy were evaluated using C-statistics of the obtained prognostic indices in the Cox regression, time-dependent receiver-operating characteristics, and decision curve analyses. Models of miRNAs, combined with relevant clinicopathological factors, were built. The five-miRNA-panel outperformed clinically established BCR scoring systems, while their combination significantly improved predictive power, based on clinicopathological factors alone. We conclude that this miRNA-based-predictor panel will be worth to be including in future studies.

## 1. Introduction

Prostate cancer (PCa) is the second most common cancer among men worldwide [1]. Six recent studies, including more than 1000 patients each, showed that approximately 15–34% of patients suffer from a biochemical recurrence (BCR), 5 to 10 years after radical prostatectomy (RP) [2,3,4,5,6,7]. The guidelines of the European Association of Urology (EAU) and the American Urological Association define BCR as a re-increase in prostate specific antigen (PSA) above 0.2 µg/L, which is confirmed by two consecutive elevated values [8,9]. This definition of BCR is associated with 90% probability of subsequently increasing PSA values [10]. At the moment, this PSA threshold is the most frequently used BCR definition in studies, and generally accepted as standard BCR value [11]. A current systematic review, based on fourteen studies which compared patients with and without BCR, confirmed BCR as decisive risk factor for distant metastasis, as well as PCa-specific and overall mortality [12]. The BCR cannot be equated with clinical recurrence after RP [9,12,13], but the clinician considers an increasing PSA level as the first sign of local recurrence or metastasis [13]. Without a secondary therapy following BCR, approximately 30% of patients experienced clinically manifested distant metastasis, and 19–27% of patients who experience BCR may suffer PCa-specific mortality within 10 years [10,14]. Early and more reliable prediction of PCa patients with high risk of BCR would aid the decision-making for adjuvant therapy, or more frequent monitoring during follow-up.

Clinicopathological variables, such as pathological tumor stage (pT stage), Gleason score and PSA, are important predictors, known to correlate with the survival of PCa patients. Currently, predictive scoring systems including these variables, are widely used for recurrence risk prediction. In 1998, D’Amico et al. [15] suggested a new risk classification method for BCR prediction in PCa patients. Subsequently, the Cancer of the Prostate Risk Assessment Postsurgical Score (CAPRAS) was used to predict BCR, based on pathological information from RP [16]. In 2005, Stephenson et al. proposed a postoperative nomogram including a number of clinicopathological variables to predict the 10-year probability of BCR after RP [17]. In addition, the National Comprehensive Cancer Network (NCCN) guidelines also suggested a novel risk classification system based on clinicopathological variables [18]. Even though clinicopathological factors certainly have a prognostic value, predictions based on them have a limited accuracy [19,20,21]. Despite the high value of the Gleason score for prognosis, there is considerable evidence that patients with the same Gleason score may experience vastly different clinical outcomes [22,23,24]. As one of the most powerful predictive scoring systems, CAPRAS was found to lack sufficient prediction accuracy in two meta-analyses [25,26]. Therefore, it is hoped that new tissue-based molecular biomarkers could be potential candidates to improve the disease recurrence prediction [27,28,29,30].

Efforts have been made to find genetic variables such as microRNAs (miRNA, miR) that may improve overall prediction accuracy [31,32,33]. We have recently reviewed all the related literature on miRNA-based BCR markers [34]. We found that more than 80% of previous studies focused only on single miRNA or multiple-miRNA-based models, while classical clinicopathological variables with known predictive value have seldom been taken into consideration during model building. However, miRNA-based predictors do not always outperform clinicopathological predictors [35]. A clinico-genomic model, combining well-defined prognostic models based on clinicopathological variables and miRNAs, should be a possibility of further improving accuracy [36,37]. Based on previous studies, it seems that models with multiple miRNAs (>2 miRNAs) perform better than a single miRNA predictor in PCa prognosis [33,38].

In this study, we aimed to (a) evaluate and compare miRNAs in prostatectomy specimens with the predictive potential of BCR reported in previous studies and filtered by a systematic literature search, (b) validate the predictive capability of these selected miRNAs by reverse-transcription quantitative real-time polymerase chain reaction (RT-qPCR), and using different evaluation approaches such as Cox regression analysis with internal validation, time-dependent receiver-operating characteristic curve analysis (ROC), and decision curve analysis, and (c) try to build a model, considering relevant clinicopathological variables combined with a multiple miRNA panel, based on the mentioned validation analyses.

## 2. Results

### 2.1. Patient Characteristics and Flow Chart of the Study

This retrospective study was performed on formalin-fixed, paraffin-embedded (FFPE) tissue specimens collected from 206 PCa patients, who underwent radical prostatectomy between 2001 and 2007 at the Charité–University Hospital. Patients were followed-up until October 2018. An alternate selection of available samples, based on patients with BCR, was conducted, so that the cohort comprised 98 (48%) patients with BCR and 108 (52%) patients without BCR. The flow diagram (Figure 1) outlines the steps performed in this study. The study was performed in accordance with the “Reporting Recommendations for Tumor Marker Prognostic Studies (REMARK)” [39].

Clinicopathological characteristics of the patient cohort, and the follow-up and recurrence free survival times, are summarized in Table 1. Statistically significant differences in most variables (age, PSA, pT status, ISUP Grade groups, and surgical margin; details in Table 1) were observed between recurrence-related PCa subgroups.

### 2.2. Selection of BCR-Related miRNAs and Their Differential Expression between Tumor and Normal Tissue and between the Samples from Patients with and without BCR

We decided to include the 20 miRNAs listed in Table 2 in our study. This selection was based on a comprehensive literature search, that we summarized in a systematic review on potential miRNA candidates of BCR prediction [34]. That review was based on 53 studies published between 2008 and 2017, and resulted in 58 distinct miRNAs as potential BCR markers. Sixteen out of these 58 miRNAs were recommended at least in two studies, or in a study with external validation, as well as partly supported by other study data when we carried out our analyses. The selection of four miRNAs (miR-31-5p, miR-34a-5p, miR-204-5p, miR-494-3p, and miR-939-5p), out of the 20 miRNAs listed in Table 2, was based on the subsequent literature data after the publication of the review [34], own data of previous microarray analyses, and/or their functional evidence regarding prostate cancer progression.

Expressions of the selected 20 miRNAs were then analyzed by RT-qPCR in 206 PCa specimens and 29 adjacent normal tissue samples. We identified 15 miRNAs, out of the 20 examined miRNAs, with differential expression in PCa patients, compared to adjacent normal controls (Appendix A; significances additionally indicated in Table 2). Of those, the expression of 10 miRNAs were found to be increased in PCa, whereas five miRNAs showed reduced levels. The expression data between the PCa specimen and adjacent normal tissue samples served only for controls and will not be further discussed. The comparison of miRNA expression in tumor tissue samples from patients with and without BCR is shown in Figure 2, and additionally included in Table 2. Significantly differential expression (*p* < 0.05) was observed for eight miRNAs, with levels always being downregulated in tissue samples from BCR patients, compared with expression levels in tissue samples from patients without BCR.

### 2.3. Correlations of Clinicopathological Variables with miRNAs and Correlations between miRNAs

Correlations between the various clinicopathological variables and the miRNAs were calculated by Spearman rank correlation analysis (Table 3).

PSA, surgical margin status, pN status, pT stage, and ISUP grade were significantly correlated with five, six, six, two, and nine different miRNAs, out of 20 examined miRNAs, respectively, while no correlations were found between age and DRE status (Table 3). For the dichotomous variables DRE, surgical margin, and pN status, and the ordinal scale variables pT stage and ISUP grade, additional calculations using the Mann–Whitney test and the Kruskal–Wallis test were compiled (Appendix A). In addition, Appendix A illustrate the behavior of the different miRNAs, depending on the pT and ISUP scale. However, all statistically significant correlations between clinicopathological variables and miRNAs are small and did not exceed the low correlation coefficient level of 0.30, with one exception (miR-1-3p and ISUP; r_s_ = −0.351). In addition, significant associations were not observed between all clinicopathological factors and differentially expressed miRNAs with regard to BCR status (Fisher’s exact test, *p*-values between 0.161 and 1.000).

Correlations between the 20 analyzed miRNAs are summarized in Appendix A. Approximately 25% of all Spearman rank correlation coefficients were higher than 0.50. However, all analyzed miRNAs were not clustered to other miRNAs forming the analyzed miRNA panel in this study, and were not related to other miRNAs of the miRNA gene family (Appendix A).

### 2.4. Prognostic Potential of miRNAs Predicting BCR

Given the mostly weak or absent correlation of the differentially expressed miRNAs with clinicopathological variables, we hypothesized that our panel of miRNA might function as useful additional markers to predict BCR. According to particular recommendation of the REMARK guidelines [39], the continuous miRNA expression data and uncategorized data were used for all further miRNA analyses to avoid reduced power in detecting associations between miRNAs and BCR prediction. Univariate Cox regression analysis was performed to evaluate the predictive capability of all 20 examined miRNAs in tumor samples (Table 4). Those 16 miRNAs with *p*-values < 0.2 were selected for multivariate Cox regression analysis to avoid type II errors (Table 4). MiRNAs that were verified as independent factors (*p* < 0.05) in subsequent multivariate Cox regression analysis in a full model, and after stepwise backward elimination, are shown in Table 4. Six miRNAs (miR-30c-5p, miR-30d-5p, miR-31-5p, miR-141-3p, miR-148a-3p, and miR-221-3p) with *p* < 0.05, except for miR-30d-5p with *p* = 0.075, remained in the final model after backward elimination. In the following, this miRNA signature was named six-miR-panel.

Furthermore, we used the split-sample method as additional internal validation method splitting the patient cohort into a training (*n* = 140) and validation (*n* = 66) set. Multivariate Cox regression analysis, with stepwise backward elimination in the training set, resulted in a model with the same five miRNAs mentioned above, without miR-30d-5p, and was named five-miR-panel (Appendix A). Using the Cox regression, we tested the predictive capacity of this five-miR-panel and the above-mentioned six-miR-panel in the total cohort, as well in the training and the test set (Table 5). C-statistics of the prognostic indices from both panels did not differ significantly, confirming that miR-30d-5p does not add a significant prediction advantage (Table 5). This was also confirmed in the decision curve analysis, showing similar curves for the five- and six-miR-panel (Appendix A). For practical reasons, we considered the five-miR-panel in further analyses.

Only two miRNAs (miR-31-5p and miR-221-3p) of this final five-miR-panel overlap with the eight miRNAs that showed significantly differential expression according to BCR status (*p* < 0.05, Table 2). Taking into account a *p*-value threshold of <0.100 with regard to the differential expression according to BCR status, four miRNAs (miR-31-5p, miR-221-3p, miR-30c-5p, and miR-141-3p) of this panel coincide with the respective 10 miRNAs (Table 2).

### 2.5. 5-miR-Panel Outperformed Models Based only on Clinicopathological Variables in Predicting BCR

Models based on clinicopathological variables are the conventional tools that are used for BCR in clinical practice to date. We therefore built models containing both clinicopathological information and miRNAs, to evaluate whether miRNAs might add a significant benefit to the prediction models based on clinicopathological factors.

In a first step, univariate Cox regression analysis of all clinicopathological variables was performed, resulting in six variables with *p* < 0.2, which were selected for further multivariate analyses (Table 6). The “full model” with all six clinicopathological factors and a “reduced model”, after stepwise backward elimination, finally including only pT stage and ISUP group grade as remaining independent predictors in the model, were established. C-statistics between the reduced and full model did not differ significantly (0.712 vs. 0.723, *p* = 0.230; Table 7).

C-statistics of the obtained prognostic indices were performed, to compare their discriminative abilities with those of other established predictive reference models mentioned in the Introduction, like CAPRAS [16], NCCN [18], according to D’Amico et al. [15] and Stephenson et al. [17] (Table 7).

As shown in Table 7, the CAPRAS model demonstrated the best AUC value among the four reference models, but its AUC value did not significantly differ from the values of our full and reduced models, (*p* = 0.187 and 0.618, respectively). We then added the five-miR-panel to all six clinicopathological models, to test whether this miRNA panel was able to improve BCR prediction significantly (Table 7). The discriminative ability of all six models based on clinicopathological variables could be significantly improved by the addition of the five-miR-panel (*p*-values of <0.0001 to 0.017).

To confirm the predictive benefit of all models, time-dependent ROC curve analyses were performed based on different postoperative time points (Figure 3). Depending on the median recurrence-free survival time of the total cohort after RP, the postoperative time points of 1, 2, 3, 4, and 5 years were applied. As shown in Figure 3a, the AUCs of the five-miR-panel and six-miR-panel were consistently higher than those of all six clinicopathological variable-based prediction models. The AUC lines of our full and reduced model, as well as the CAPRAS model, were quite similar, and consistently higher than those of the three remaining reference models at all time points. However, as shown in Figure 3b, inclusion of the five-miR-panel into the clinicopathological variable-based models improved all AUCs, which is consistent with the results in Table 7.

We also applied decision curve analysis as recommended most informative metrics [78] to demonstrate the incremental prognostic value of the five-miR-panel. It is exemplarily shown in the curves of Figure 4 for the reference models according to D’Amico et al. [15] and CAPRAS [16], as well as for our “full model”, in comparison to those of the five-miR-panel and the combined models. The curves indicate the benefit of the combined models in comparison to both single models, whereas C-statistics is not able to detect differences between the results of the combined models and the five-miR-panel (Appendix A).

### 2.6. Functional Links Between miRNAs of the 5-miR-Panel and Prostate Cancer

Validated functional links between the miRNAs of the five-miR-panel and PCa are given in Table 8. Both tumor suppressive and oncogenic effects were observed for miR-141-3p, miR-148a-3p, and miR-221-3p. The differences of the five miRNAs, with regard to their expression levels (Table 2), hazard ratios (Table 4), and tumor suppressive/oncogenic functions (Table 8), show that associations between the molecular mechanisms and the outcome results are not evident. This is not surprising, because every miRNA can affect numerous target genes in different ways, whereas outcome results, here as predicted BCRs, are related to the integration of several data.

## 3. Discussion

In this retrospective study, we evaluated the potential of 20 individual miRNAs alone, or in combinations with clinicopathological variables, to predict BCR after RP. As shown in the present study, five miRNAs (miR-30c-5p, miR-31-5p, miR-141-3p, miR-148a-3p, and miR-221-3p) were finally validated as independent prognostic panel of BCR. Thus, the five-miR panel, together with standard clinicopathological factors, is a novel promising predictor of BCR, and will aid in decision-making and treatment management of PCa patients at risk of recurrence after RP.

As briefly outlined in the Introduction, the present study was designed to avoid the numerous shortcomings of previous studies, that we characterized in a systematic review of studies that used microRNAs as BCR predictors after RP [34]. For example, 44 out of the evaluated 53 studies (83%) measured expression levels of only one or two miRNAs; 18 studies (34%) calculated the predictive ability of miRNAs only in univariate analyses, without considering clinicopathological variables; and only nine studies (17%) showed an improved BCR prediction ability of miRNAs, in comparison to the prediction ability with clinicopathological variables; only 10 studies (19%) included a BCR group with more than 30 patients; and internal/external validation of the predictive models using miRNAs was reported in only eight studies (15%) [34]. Therefore, the present study was prepared with a total of 206 patients, including 98 patients suffering from BCR, and determination of 20 miRNAs. As these miRNAs were already proven in other studies as potential BCR predictors (Table 2), this approach provided a good opportunity to overcome the deficiencies of the previously reviewed miRNA-oriented BCR prediction studies. In this respect, two objectives were outlined in the present study, namely to (i) generate a robust multiple miRNA-based classifier and (ii) combine this classifier with relevant standard clinicopathological variables to a reliable model [36,91]. Nam et al. [38] described a panel of five miRNAs as a novel and promising predictor, after adjustment and combination with known clinicopathological variables. In addition, Kristensen et al. [33] showed that the prognostic benefit of a model with clinicopathological variables could be improved by the inclusion of three miRNAs. Both studies provide good examples for the establishment of clinico-genomic prediction models, based on miRNA classifier with multiple miRNAs.

By multivariate Cox regression analyses, we developed a six-miR-panel and five-miR-panel as effective BCR predictor classifiers. Two independent internal validation approaches, bootstrapping and the split-sample method, confirmed the robustness of these models, in line with the meaning of a “fit-for purpose” method [92]. C-statistics and decision curve analysis of both panels indicated similar performance for both panels (Table 5, Appendix A). We therefore decided to use the five-miR-panel as the final classifier for further analyses, in accordance with the principle “do not utilize more laboratory tests than absolutely necessary” [93].

The predictive potential of the individual miRNAs included in our final five-miR-panel was reported in previous studies (Table 2). For instance, downregulated expression of miR-30c-5p was an independent BCR marker in multivariate regression analysis including clinicopathological variables, [50] and reduced levels of miR-30c-5p were also found to be significantly associated with Gleason score and pathological stage [50], an association that was not confirmed in our study. In addition, downregulated miR-30c-5p expression, combined with expression of its target gene BCL9, was considered an independent predictor in multivariate regression analysis in another study [51]. Five previous studies focused on the role of miR-221-3p in predicting BCR [33,45,46,59,67]. Three of these studies confirmed reduced miR-221-3p levels to be a promising predictive biomarker, using multivariate Cox regression analysis, with adjustment of clinicopathological variables [33,45,67]. Furthermore, a strong correlation was found in these three studies between miR-221-3p expression, pathological stage, Gleason score, and pre-operative PSA levels, which was also not confirmed in our study. Two studies failed to detect the predictive potential of miR-221-3p in PCa patients in multivariate analyses, which may have resulted from the short follow-up time (median <2 years) as well as analytical reasons [46,59]. In two other studies [45,47], downregulated miR-141-3p was found to be associated with BCR, but validation as an independent predictor failed in multivariate regression analysis. Reduced levels of miR-148a-3p were observed to correlate with BCR in the study of Lichner et al. [47]. MiR-31-5p was already described as a deregulated miRNA in PCa [54,94] and reported in relationship to BCR [55], but was confirmed as a predictive biomarker of BCR in our multivariate prediction model for the first time.

We found important particularities between our study and the studies by Nam et al. [38] and Kristensen et al. [33]. Nam et al. [38] used five miRNAs in their signature, including miR-301a, which was most strongly associated with PCa recurrence. This miRNA was also included in our initial 20 miR measurement panel, but was not statistically significant when associated with BCR in univariate Cox regression analysis (*p* = 0.246, Table 4). The multiple miRNA signature by Kristensen et al. [33] consists of three miRNAs: miR-185-5p, miR-221-3p, and miR-326. All three miRNAs were also used in our initial panel and were significantly associated with BCR (Table 4), but only miR-221-3p remained as an independent marker in the final panel of multivariate Cox regression analysis after stepwise backward elimination (Table 4). Thus, it is worth noting that significant miRNAs in the other signatures failed in their inclusion in our final panel. After we had finished the experimental work in our study, three other panels for prediction of PCa outcome after RP based on multiple miRNA have been published. The working group of Sorensen et al. [95,96], with the above-mentioned panel of three miRNAs, introduced two novel panels. These are a four-miRNA ratio (miR-23a-3p x miR-10b-5p/miR-133a x miR-374b-5p), named MICaP [95], and a panel based on nine miRNAs, among them miR-185-5p and miR-221-3p, combined with three methylation markers, named miMe [96]. Nam et al. [97] introduced a new five-miRNA panel associated with metastasis after RP, while no miRNA was identical with those miRNAs recommended in their preceding report [38]. All these new panels showed promising predictive capacities, but the panels differed and were not compared within the same working group. Consistently good prediction accuracy data under conditions of differing miRNAs in the panels may be the result of different starting conditions, in particular the initial measurements of miRNAs, and the multivariate assessment of data. Ultimately, it is scarcely surprising that our five-miR-panel contains only two miRNAs with significantly different expression levels (*p* < 0.05) between the two BCR groups and three other miRNAs with p-values > 0.05. This panel is the result of multivariate data assessment that considers possible interactions between the various factors. On the other hand, it additionally confirms our selection approach of including all miRNAs in multivariate Cox regression analysis that had *p*-values < 0.200 in univariate Cox regressions. This procedure enables us to avoid type II errors, because all five miRNAs show characteristics of independent factors (Table 4).

The missing or weak correlations/associations of individual miRNAs of the final five-miR-panel, with conventional clinicopathological variables and among each other, characterize their potential as orthogonal predictors (Table 3, Appendix A) [98]. Orthogonal biomarkers are characterized by the fact that they can improve the prediction accuracy of BCR, due to their additional information, independently from other variables. Associations of miRNAs with clinicopathological variables, for example, ISUP grading with miR-31-5p, miR-141-3p, and miR-148a-3p, did not exclude their potential as common independent predictive BCR biomarkers (Table 3, Appendix A) [47].

Until now, the BCR prediction and other outcomes in clinical practice were generally based on prediction tools using standard clinicopathological variables (Table 7). In addition, a current systematic review of the prognostic impact of BCR on oncological outcome considered only clinical factors and tumor characteristics [12]. The limited prognostic accuracy of different approaches has been critically discussed [19,20,21,26]. Recently, several tissue-based tests known as Decipher, OncoType DX Prostate, Prolaris, and ProMark, using multigene or multiprotein expression data, have been introduced into clinical practice for PCa risk stratification and prognosis (reviewed in [99]). The NCCN Prostate Cancer Guidelines Panel stated in their 2019 guidelines that these tests might be considered for an initial risk stratification, based on biopsy samples and RP specimens [99]. Other current reports support this view [27,28,29,30,100,101,102]. Yet, the particularly different prognostic outcome results obtained by a head-to-head comparison of three tests (namely Decipher, Prolaris, and Oncotype), obviously complicate the routine implementation of these tests [103]. MiRNA-based panels, like the here presented five-miR-panel, have a great advantage in comparison to the mentioned multigene tests, because miRNAs are more robust molecular analytes than mRNAs [104]. This is particularly important under clinical routine conditions to procure stable tissue samples.

We would also classify the analysis of molecular biomarkers (specifically miRNA expression) in tissue specimens, either FFPE or fresh-frozen samples, from RP, as a promising tool to improve the prognostic accuracy of clinicopathological factors only. C-statistics and decision curve analysis clearly indicates that the five-miR-panel significantly improved the predictive accuracy of all clinicopathological models. Even the CAPRAS tool, which outperformed all other clinical models, was significantly improved. Our results indicate that there is considerable potential for improvement of the predictive ability and accuracy of the currently existing prognostic models (Table 7, Figure 3 and Figure 4). The mentioned studies by Nam et al. [38] and Kristensen et al. [33] support this view. We want to add that the sole focus on molecular markers to improve the prognostic accuracy contradicts the necessary global view in translational medicine to consider all aspects of disease associated features appropriately [105].

Besides identifying novel biomarkers, the identification of dysregulated miRNAs can also help improve understanding of the mechanisms of prostate cancer recurrence, which may provide a basis for novel therapeutic strategies or a better understanding of current treatment response. MiRNAs can function as both oncogenes and tumor suppressors [106]. The complexity of interpreting the functional role of the miRNAs that are relevant markers for BCR prediction can be illustrated by miR-141-3p and 148-3p, both included in the five-miR-panel. MiR-141-3p was characterized as a tumor suppressor in PCa, and inhibits PCa cell proliferation and migration by repressing genes involved in extracellular matrix-mediated pathways [47]. On the other hand, miR-141-3p was also described to act as an oncogene in untreated PCa and castration-resistant PCa, by enhancing PCa cell proliferation, even though no target genes have been described [107]. MiR-148-3p was shown to act as an oncogene by promoting prostate cancer cell growth due to repression of its target CAND1 [87], but it was also described as a tumor suppressor, by inhibiting the growth of androgen-refractory PCa cells through repression of mitogen- and stress-activated protein kinase 1 [88]. Regarding the different types of prostate cancer cells used in these studies, we assume that the expression of both miR-141 and miR-148a might be continuously altered during PCa progression. Both examples highlight that it might not be meaningful to characterize specific effects of individual miRNAs without considering possible interactions between the different molecular components. MiRNAs are part of a large network of competing endogenous RNAs, which further consists of mRNAs and long non-coding RNAs, that are regulated via miRNA responsive elements [108]. Changes in this network could explain the different functional outcome of an expressed miRNA in a specific setting, like in forthcoming BCRs. Moreover, the functional impact of a biomarker might explain the biological rationale of its effect, but does not reflect its clinical applicability and validity [109]. This view is supported by the fact that a concordance between the expression levels of the BCR and non-BCR samples, the hazard ratios, and the functional mechanisms of the miRNAs of the five-miR-panel, is not evident (Table 2, Table 4 and Table 8).

Despite our effort to avoid deficiencies observed in previous studies, as discussed above, some limitations of our study should be noted. These are the retrospective nature of the study, the sample collection from only a single center, and the lack of external validation. Another limitation is the focus on BCR as an endpoint, without considering clinical endpoints like metastasis-free survival or cancer-specific-free survival. Because of partially unclear or unavailable data, corresponding analyses were not possible, but were also not primarily intended. Future studies should focus on validating the predictive power of the five-miRNA panel in a prospective and multi-institutional setting. It would be worthwhile to include a head-to-head comparison with other multiple-miRNA-based tools, using a currently published novel EAU BCR risk stratification system as outcome endpoint [110].

## 4. Materials and Methods

### 4.1. Patient Selection and Data Collection

This retrospective study was approved by the local Ethics Committee of the Charité–University Hospital (EK-CCM-2004-09-14, approval date: September 20, 2004; EA1/153/07, approval date: October 22, 2007; EA1/134/12, approval date: June 22, 2012) and informed patient consent was obtained. The study was carried out in accordance with the Declaration of Helsinki. FFPE tissue specimens were collected from 206 PCa patients who underwent RP between 2001 and 2007 at Charité–University Hospital. Patients were treatment-naive before RP and were followed-up until October 2018. Follow-up data were based on medical records, telephone contacts with the patients’ urologists and patients or family members. Sample size was determined by a power-adapted calculation (α = 5%, power = 80%; Appendix A). The cohort comprised 98 (48%) patients with BCR, and 108 (52%) patients without BCR, in the follow-up after RP (Table 1). According to the guidelines of the European Association of Urology (EAU) and the American Urological Association (AUA), BCR was defined as a re-increase of PSA above 0.2 µg/L, which is confirmed by consecutive elevated values [8,9]. Twenty-nine samples from adjacent normal tissues were used as controls.

Histopathological evaluation, grading, staging, and determination of margin status are critical parameters for prognostic considerations in PCa. Pathological specimens were reviewed by two independent pathologists (E.K., A.E), in order to provide an unbiased and unanimous pathological diagnosis for all cases. Criteria of the International Union Against Cancer and the World Health Organization/International Society of Urological Pathology (ISUP) were applied.

### 4.2. RNA Extraction and Reverse Transcription Quantitative Real-Time Polymerase Chain Reaction (RT-qPCR)

All procedures were carried out as documented in detail in our previous publications [32,54,104,111,112], taking into account the items of “Minimum Information for Publication of Quantitative Real-time PCR Experiments” (MIQE guidelines) [113]. Briefly, total RNA was extracted from dissected FFPE tissue using the miRNeasy FFPE kit (Qiagen, Hilden, Germany), according to the manufacturer’s protocol. The RNA concentration and purity were measured on a NanoDrop ND-1000 spectrophotometer (NanoDrop Technologies, Wilmington, DE, USA). RNA extracts showed absorbance ratios of 260 nm to 280 nm and of 260 nm to 230 nm between 1.84 to 2.04 and 1.71 to 1.92, respectively, confirming their appropriate purity for further measurements. The selection of the 20 potential miRNA candidates for BCR prediction in this study has been explained in the above-mentioned Section 2.2. RT-qPCR analyses were carried out on the Light Cycler 480 Instrument (Roche Diagnostics, Mannheim, Germany) using TaqMan miRNA assays (Applied Biosystems, Foster City, CA, USA; Appendix A). Expression values of all examined miRNAs were normalized to the geometric mean of let-7g-5p and miR-103a-3p, using the software qBasePLUS, v.3.0 software (Biogazelle, Zwijnaarde, Belgium). The suitability of both miRNAs as stable reference miRNAs in the normalization process is shown in Appendix A.

### 4.3. Statistical Analysis

IBM SPSS Statistics for Windows, version 25 (IBM Corp., Armonk, NY, USA) with bootstrap module, GraphPad Prism 8.20 (GraphPad Software, La Jolla, CA, USA), and MedCalc 19.06 (MedCalc Software, Ostend, Belgium) were used for statistical analysis. Nonparametric statistical tests used in our study included the Mann–Whitney U test, Chi-square or Fisher’s exact test, and Spearman rank correlation test. Univariate and multivariate Cox regression analyses were carried out for the prediction of BCR. For the internal validation of the models, bias-corrected and accelerated bootstrap calculations with the total cohort and the split-sample method (total cohort was randomly divided into a training (*n* = 140) and test (*n* = 66) set) were performed. Output data of Cox regression analyses defined as prognostic indices were utilized for C-statistics and decision curve analyses, as previously described [114,115]. AUCs under the time-dependent ROC curves of all models were additionally computed based on a cumulative case/dynamic control approach [77] at postoperative 1, 2, 3, 4, and 5 years, respectively. A *p*-value of < 0.05 (two-sided) was regarded as statistically significant.

## 5. Conclusions

In conclusion, we have identified miRNAs that are significantly associated with BCR after RP in prostate cancer, namely miR-30c-5p, miR-31-5p, miR-141-3p, miR-148a-3p and miR-221-3p. The inclusion of this five-miRNA-panel significantly improved the predictive accuracy of known clinicopathological models. Our study provides a good example of the benefit of incorporating molecular biomarkers, like miRNAs, into previously established risk prediction models based only on clinicopathological factors. Such an approach would be useful to identify patients at risk, set up personalized surveillance protocols, counsel patients, and select patients for adjuvant treatment trials. However, future prospective studies are needed, and comparison with other genomic classifiers, to implement a robust and reliable tool into the decision-making process of clinical practice.

## Figures and Tables

**Figure 1 cancers-11-01603-f001:**
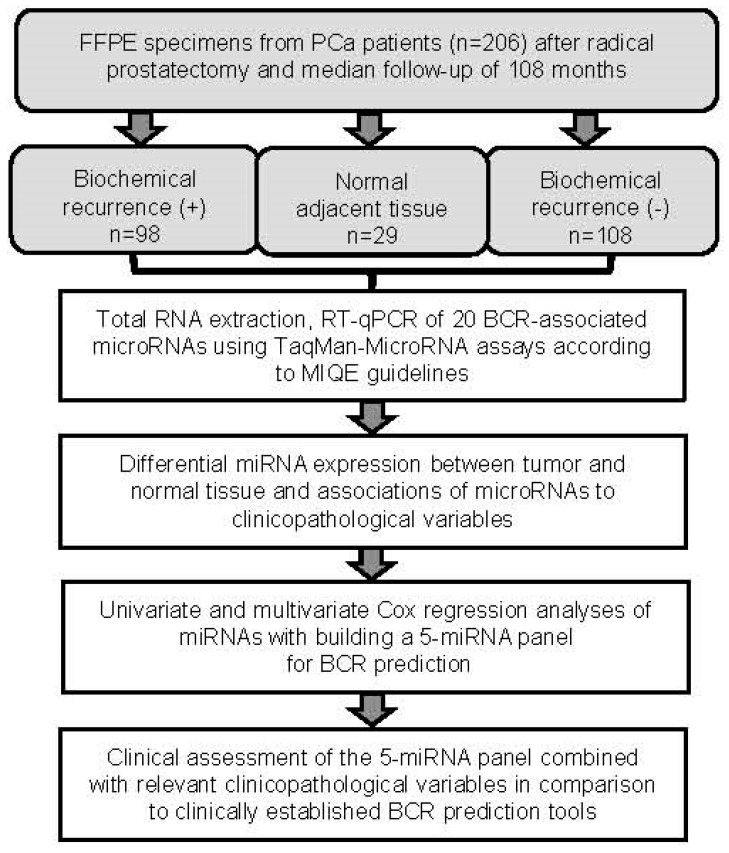
Flowchart of the study. Fourteen and 15 out of 29 adjacent normal samples were from patients without and with BCR, respectively. Between these two sample groups, expression of all microRNAs of interest in this study shown in Figure 2 did not significantly differ (Mann-Whitney *U*-test, *p*-values between 0.198 and 0.983) and the combined 29 samples were therefore used as “Normal adjacent tissue”. Abbreviations: FFPE, formalin-fixed, paraffin-embedded; PCa, prostate carcinoma; BCR, biochemical recurrence; RT-qPCR, reverse transcription real time quantitative polymerase chain reaction; MIQE, Minimum Information for Publication of Quantitative Real-time PCR Experiments.

**Figure 2 cancers-11-01603-f002:**
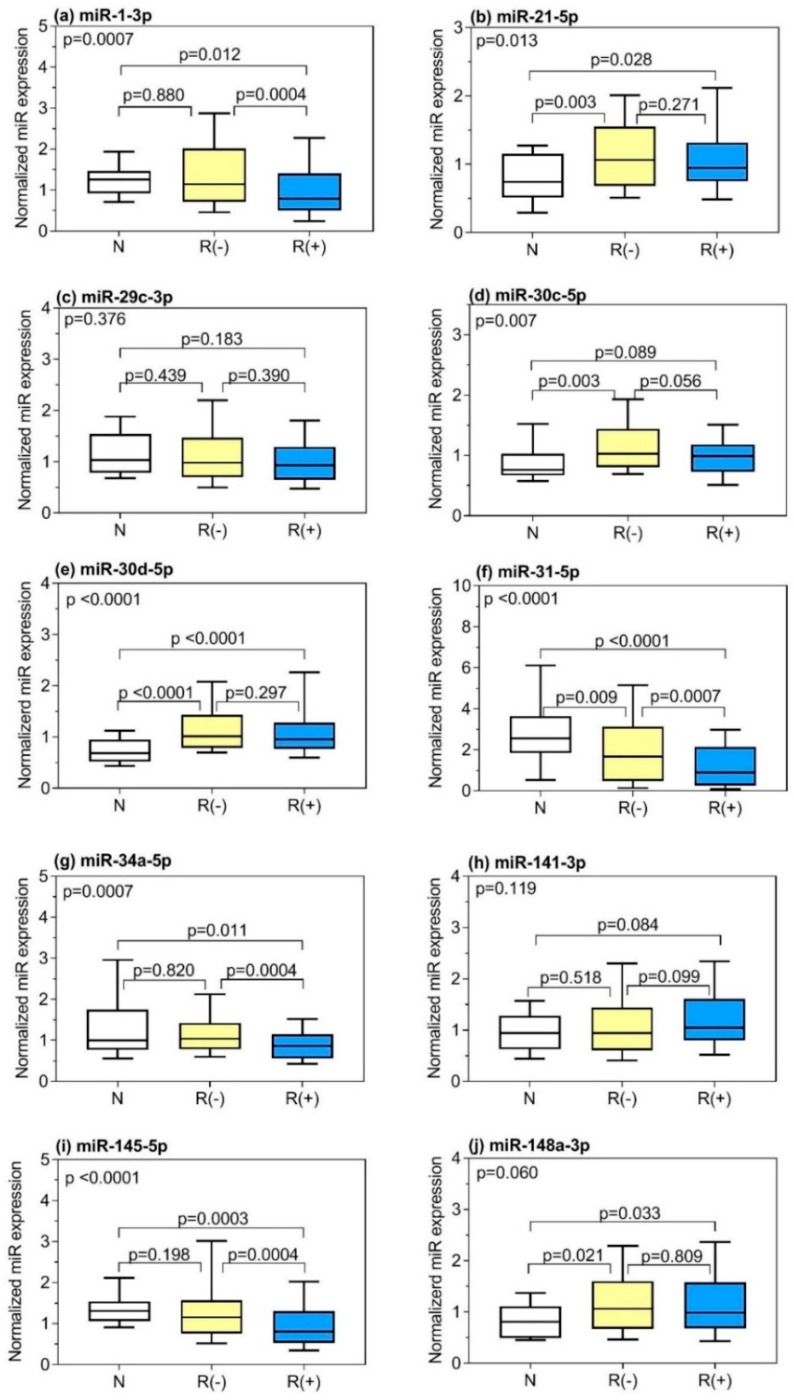
Expression of 20 microRNAs examined in adjacent normal tissue samples and prostate cancer (PCa) tissue samples from patients with and without biochemical recurrence (BCR) are presented in subfigures (**a**–**t**). Normalized miRNA expression data are given as box- and whisker plots. Boxes represent the lower and upper quartiles with medians; whiskers illustrate the 10 and 90 percentiles of the cohorts. Kruskal–Wallis test with Dunn’s post-hoc was performed. The *p*-value of the total Kruskal–Wallis test is indicated in the upper left-hand corner of the figure. Abbreviations: N, adjacent normal tissue (*n* = 29) as explained in legend of Figure 1; R (−), PCa tissue samples from patients without BCR (*n* = 108); R (+), PCa tissue samples from patients with BCR (*n* = 98).

**Figure 3 cancers-11-01603-f003:**
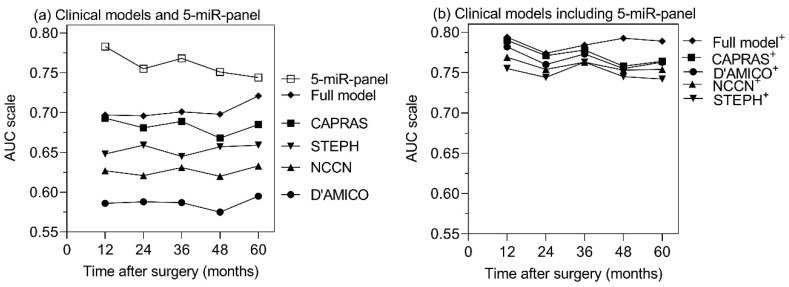
Time-dependent receiver-operating characteristics (ROC) curve analyses of (**a**) separate and (**b**) combined miRNA panel-based and clinicopathological factors-based models at different postoperative time points. Areas under the time-dependent ROC curve (AUCs) of all models were computed based on a cumulative case/dynamic control approach [77] at postoperative 1, 2, 3, 4, and 5 years, respectively. The models of miRNA-panels are described in Table 4 and Appendix A. The four reference clinical models [15,16,17,18] and our full model are explained in Table 6 and Table 7. Abbreviations: CAPRAS, Cancer of the Prostate Risk Assessment Postsurgical Score, calculated according to [16]; STEPH, calculated according to Stephenson et al. [17]; NCCN, National Comprehensive Cancer Network, calculated according to [18]; D’AMICO, calculated according to D’Amico et al. [15]; Full model, according to the Cox regression model described in Table 6 with all clinicopathological factors except for digital rectal examination.

**Figure 4 cancers-11-01603-f004:**
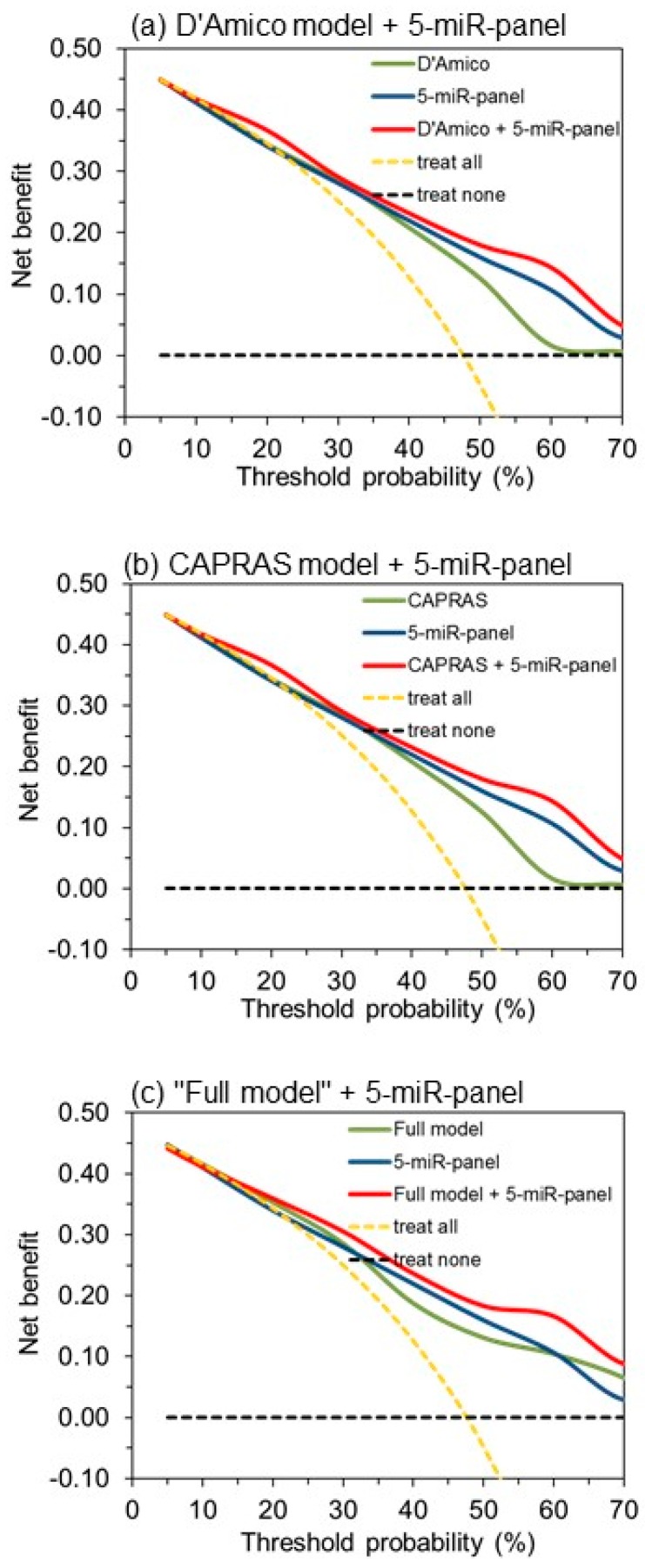
Decision curve analysis to demonstrate the benefit of the inclusion of the five-miR-panel into only clinicopathological variables-based biochemical recurrence predictive tools, according to (**a**) D’Amico et al. [15], (**b**) Cancer of the Prostate Risk Assessment Postsurgical Score (CAPRAS) [16], and (**c**) the “full model” used in this study (Table 6).

**Table 1 cancers-11-01603-t001:** Clinicopathological characteristics of the study group.

Characteristics	All Patients	Patients with Biochemical Recurrence	Patients without Biochemical Recurrence	*p*-value ^a^
Patients, no. (%)	206 (100)	98 (48)	108 (52)	
Age, median years (range)	63 (47–74)	64 (51–74)	62 (47–73)	0.014
PSA, median µg/L (range)	8.4 (1.3–50.6)	9.3 (1.4–50.6)	7.5 (1.3–32.9)	0.003
Prostate volume, median cm^3^ (range)	33 (14–130)	33 (15–130)	32 (14–120)	0.465
DRE, no. (%)				0.260
Non-suspicious	122 (59)	54 (55)	68 (63)	
Suspicious	84 (41)	44 (45)	40 (37)	
pT status, no. (%)				<0.0001
pT2a	18 (9)	5 (5)	13 (12)	
pT2b	28 (14)	6 (6)	22 (20)	
pT2c	76 (37)	31 (32)	45 (42)	
pT3a	62 (30)	40 (41)	22 (20)	
pT3b	21 (10)	15 (15)	6 (6)	
pT4	1 (0.5)	1 (1)	0	
ISUP Grade groups, no. (%)				
1	52 (25)	14 (14)	38 (35)	0.0003
2	68 (33)	30 (31)	38 (35)	
3	27 (13)	19 (19)	8 (7)	
4	29 (14)	16 (16)	13 (12)	
5	30 (15)	19 (19)	11 (10)	
pN status, no. (%) ^b^				
pN0	113 (55)	57 (58)	56 (52)	0.207
pN1	10 (5)	7 (7)	3 (3)	
pNx	83 (40)	34 (35)	49 (45)	
Surgical margin, no. (%)				
Negative	133 (65)	54 (55)	79 (73)	0.009
Positive	73 (35)	44 (45)	29 (27)	
Follow-up after surgery				
Median months (range)	108 (17–180)	101 (19–160)	121 (17–180)	<0.0001
Recurrence free survival				
Median months (95% CI)	52 (46–60)	16 (11-21)	80 (75–93)	<0.0001

Abbreviations: CI, confidence interval; DRE, digital rectal examination; ISUP Grade groups, histopathological grade system based on Gleason score according to the International Society of Urologic Pathology; pN, lymph node status; PSA, prostate specific antigen; pT, pathological tumor classification. ^a^
*p*-values (Mann-Whitney U test; Chi-square Cochran-Armitage test for trend or Fisher’s exact test) indicate the association of the patients with and without biochemical recurrence with the clinicopathological variables. ^b^ In the following evaluation of data, with regard to the pN status, patients with pNx (with no lymph node dissection) were considered node negative, and were combined with pN0 patients into one group.

**Table 2 cancers-11-01603-t002:** MicroRNAs analyzed in this study. The selection of microRNAs was based on a previous systematic review of studies with microRNAs as potential BCR predictors after RP [34] and/or on other study data after the publication of the review.

miRBase ID Release 22	miRBase Accession No. ^a^	Reference to BCR-Related miRNA	Differential Expression: Tumor vs. Normal Tissue ^b^	Differential Expression: Recurrence vs. Non-Recurrence ^c^
hsa-miR-1-3p	MIMAT0000416	[35,40,41]	n.s., *p* = 0.177	↓, *p* = 0.0004
hsa-miR-21-5p	MIMAT0000076	[42,43,44,45,46]	↑, *p* = 0.006	n.s., *p* = 0.271
hsa-miR-29c-3p	MIMAT0000681	[47,48,49]	n.s., *p* = 0.272	n.s., *p* = 0.390
hsa-miR-30c-5p	MIMAT0000244	[50,51]	↑, *p* = 0.011	n.s., *p* = 0.056
hsa-miR-30d-5p	MIMAT0000245	[52,53]	↑, *p* < 0.0001	n.s., *p* = 0.297
hsa-miR-31-5p	MIMAT0000089	[54,55]	↓, *p* < 0.0001	↓, *p* = 0.0007
hsa-miR-34a-5p	MIMAT0000255	[47,56,57]	n.s., *p* = 0.155	↓, *p* = 0.0004
hsa-miR-141-3p	MIMAT0000432	[45,47]	n.s., *p* = 0.219	n.s., *p* = 0.099
hsa-miR-145-5p	MIMAT0000437	[58,59,60,61]	↓, *p* = 0.011	↓, *p* = 0.0004
hsa-miR-148a-3p	MIMAT0000243	[47,62]	↑, *p* = 0.018	n.s., *p* = 0.809
hsa-miR-185-5p	MIMAT0000455	[33]	↑, *p* = 0.015	n.s., *p* = 0.420
hsa-miR-195-5p	MIMAT0000461	[63,64]	↑, *p* < 0.0001	n.s., *p* = 0.559
hsa-miR-204-5p	MIMAT0000265	[65,66]	↓, *p* = 0.0001	↓, *p* = 0.045
hsa-miR-221-3p	MIMAT0000278	[33,45,59,60,67]	↓, *p* < 0.0001	↓, *p* = 0.0001
hsa-miR-224-5p ^d^	MIMAT0000281	[68,69]	↓, *p* < 0.0001	↓, *p* = 0.0001
hsa-miR-301a-3p	MIMAT0000688	[38,70]	↑, *p* = 0.0002	n.s., *p* = 0.674
hsa-miR-326	MIMAT0000756	[33]	↑, *p* = 0.009	↓, *p* = 0.033
hsa-miR-374b-5p	MIMAT0004955	[71,72]	↑, *p* < 0.0001	n.s., *p* = 0.141
hsa-miR-494-3p	MIMAT0002816	[73,74,75]	n.s., *p* = 0.721	n.s., *p* = 0.821
hsa-miR-939-5p	MIMAT0004982	[76]	↑, *p* = 0.028	n.s., *p* = 0.784

Abbreviations: BCR, biochemical recurrence; RP, radical prostatectomy; HR, hazard ratio; ↑, upregulated; ↓, downregulated; n.s., not significant. ^a^ Further details about the characteristics of the analyzed microRNAs (location on chromosomes, miRNA families, clustering with other miRNA, and assays) are summarized in Appendix A. ^b^ Expression difference (Mann–Whitney test) in all tumor samples (*n* = 206) vs. controls of adjacent normal tissue samples as controls (*n* = 29), see also Appendix A. ^c^ Expression difference (Kruskal–Wallis test with Dunn’s post-hoc test, see also Figure 2) in tissue samples from BCR patients vs. those from non-BCR patients. ^d^ Until miRBase 21; however, this miRNA has now the miRBase ID eca-miR-224 (accession no. MIMAT0013206).

**Table 3 cancers-11-01603-t003:** Spearman rank correlation coefficients between various clinicopathological factors and the examined microRNAs in all tumor samples (*n* = 206). The eight significantly differentially expressed miRNAs, according to biochemical recurrence status, and correlations with *p* < 0.05, are highlighted in gray.

**(A) miR-1 to miR-1 to miR148a**											
		**miR-1**	**miR-21**	**miR-29c**	**miR-30c**	**miR-30d**	**miR-31**	**miR-34a**	**miR-141**	**miR-145**	**miR-148a**
**Age**	r_s_	−0.059	−0.087	0.104	0.009	0.073	−0.094	0.070	0.095	−0.036	0.042
	*p*-value	0.403	0.214	0.139	0.901	0.294	0.178	0.318	0.174	0.604	0.546
**PSA**	r_s_	−0.218	0.007	0.030	−0.107	0.019	−0.086	−0.119	0.156	−0.211	0.068
	*p*-value	0.002	0.922	0.668	0.125	0.783	0.220	0.090	0.025	0.002	0.334
**DRE**	r_s_	−0.119	0.132	−0.035	−0.127	0.021	−0.053	−0.031	−0.022	−0.062	0.005
	*p*-value	0.087	0.059	0.622	0.068	0.765	0.446	0.657	0.757	0.376	0.945
**Margin**	r_s_	−0.146	−0.129	0.056	−0.139	−0.064	−0.198	−0.114	0.112	−0.184	0.184
	*p*-value	0.037	0.065	0.425	0.047	0.358	0.004	0.102	0.108	0.008	0.008
**pN status**	r_s_	−0.110	0.022	−0.147	−0.122	−0.091	−0.129	−0.142	−0.046	−0.110	0.005
	*p*-value	0.116	0.753	0.035	0.080	0.194	0.064	0.041	0.508	0.114	0.946
**pT stage**	r_s_	−0.218	0.146	0.029	−0.086	0.015	−0.085	−0.061	0.153	−0.179	0.144
	*p*-value	0.002	0.036	0.677	0.222	0.828	0.225	0.385	0.028	0.010	0.038
**ISUP**	r_s_	−0.351	0.198	0.188	−0.080	0.095	−0.232	−0.014	0.298	−0.265	0.249
	*p*-value	0.0001	0.004	0.007	0.256	0.173	0.001	0.844	0.0001	0.0001	0.0003
**(B) miR-185 to miR-939**											
		**miR-185**	**miR-195**	**miR-204**	**miR-221**	**miR-224**	**miR-301a**	**miR-326**	**miR-374b**	**miR-494**	**miR-939**
**Age**	r_s_	−0.002	0.011	0.119	0.001	−0.088	−0.044	0.045	−0.036	0.038	0.000
	*p*-value	0.976	0.870	0.090	0.983	0.208	0.528	0.517	0.611	0.584	0.999
**PSA**	r_s_	0.063	−0.137	−0.220	−0.080	−0.137	−0.030	0.048	−0.063	0.005	−0.027
	*p*-value	0.371	0.050	0.002	0.253	0.049	0.670	0.495	0.367	0.945	0.702
**DRE**	r_s_	0.012	0.033	−0.062	−0.082	−0.038	0.001	0.000	−0.034	0.071	−0.003
	*p*-value	0.860	0.634	0.373	0.244	0.588	0.990	0.994	0.632	0.311	0.966
**Margin**	r_s_	−0.096	−0.214	−0.124	−0.032	−0.175	−0.080	−0.092	−0.096	0.127	−0.030
	*p*-value	0.169	0.002	0.076	0.652	0.012	0.254	0.186	0.168	0.069	0.672
**pN status**	r_s_	−0.020	0.031	−0.110	−0.048	−0.106	0.034	−0.080	−0.097	−0.031	−0.130
	*p*-value	0.774	0.659	0.116	0.496	0.131	0.631	0.256	0.166	0.659	0.062
**pT stage**	r_s_	0.088	0.032	−0.100	−0.083	−0.166	0.121	−0.023	−0.064	−0.049	−0.065
	*p*-value	0.211	0.648	0.154	0.235	0.017	0.084	0.738	0.361	0.485	0.352
**ISUP**	r_s_	0.092	−0.042	−0.256	−0.015	−0.146	0.051	0.066	−0.059	0.056	−0.069
	*p*-value	0.186	0.551	0.0002	0.825	0.036	0.467	0.346	0.402	0.423	0.323

Abbreviations: miR, microRNA in its abbreviated form to facilitate the readability of the table; full annotation in Table 2; r_s_, Spearman rank correlation coefficient; PSA, prostate specific antigen; DRE, digital rectal examination result; pN status, pathological lymph node status, positive/negative; pT stage, pathological tumor classification, see Table 1; ISUP, histopathological grade system, see Table 1.

**Table 4 cancers-11-01603-t004:** Construction of a miRNA-based predictive classifier for biochemical recurrence, using a bootstrapping approach of Cox regression analysis with all 206 tumor samples. ^a^

miRNA	Univariate Cox Regression for All miRs		Multivariate Cox Regression with Significant Univariate miRs ^b^			
			Full Model		Backward Elimination	
	HR (95% CI)	*p*-value	HR (95% CI)	*p*-value	HR (95% CI)	*p*-value
miR-1-3p	0.67 (0.52–0.87)	0.003	0.82 (0.58–1.16)	0.262		
miR-21-5p	1.12 (1.01–1.24)	0.049	1.35 (0.86–2.12)	0.188		
miR-29c-3p	1.06 (0.81–1.39)	0.660				
miR-30c-5p	0.56 (0.34–0.92)	0.023	0.46 (0.23–0.93)	0.031	0.49 (0.28–0.85)	0.011
miR-30d-5p	1.22 (1.05–1.42)	0.009	1.79 (1.12–2.85)	0.015	1.27 (0.98–1.66)	0.075
miR-31-5p	0.78 (0.68–0.90)	0.001	0.83 (0.69–0.98)	0.030	0.78 (0.67–0.91)	0.001
miR-34a-5p	1.09 (0.96–1.23)	0.174	0.61 (0.36–1.03)	0.063		
miR-141-3p	1.25 (1.10–1.41)	0.001	1.96 (1.25–3.07)	0.003	1.92 (1.32–2.79)	0.001
miR-145-5p	0.68 (0.51–0.90)	0.008	1.17 (0.65–2.11)	0.604		
miR-148a-3p	1.15 (0.99–1.34)	0.064	0.58 (0.35–0.95)	0.031	0.60 (0.44–0.81)	0.001
miR-185-5p	1.06 (1.01–1.12)	0.024	0.99 (0.68–1.43)	0.955		
miR-195-5p	0.91 (0.68–1.23)	0.553				
miR-204-5p	0.76 (0.59–0.98)	0.033	1.31 (0.92–1.88)	0.138		
miR-221-3p	0.81 (0.67–0.98)	0.033	0.67 (0.48-0.95)	0.024	0.74 (0.61–0.90)	0.002
miR-224-5p	0.67 (0.50–0.89)	0.006	0.79 (0.51–1.22)	0.280		
miR-301a-3p	0.83 (0.61–1.14)	0.241				
miR-326	1.03 (1.01–1.05)	0.008	0.97 (0.80–1.18)	0.749		
miR-374b-5p	0.85 (0.52–1,38)	0.509				
miR-494-3p	1.00 (1.00–1.01)	0.023	1.01 (0.99–1.04)	0.447		
miR-939-5p	1.02 (1.00–1.05)	0.021	0.98 (0.80–1.21)	0.869		

Abbreviations: HR, hazard ratio; CI, confidence interval. ^a^ Ninety-nine patients with biochemical recurrence and 108 patients without biochemical recurrence. ^b^ The 16 miRNAs with *p*-values < 0.200 in univariate Cox regression analysis were used in the Full model of multivariate Cox regression analysis, and the six miRNAs with *p*-values < 0.05 in this Full model were subsequently used in a Backward elimination approach.

**Table 5 cancers-11-01603-t005:** Comparison of the C-statistics of prognostic indices for BCR prediction using two miRNA panels in Cox regression analyses.

miRNA Panel	All Samples (*n* = 206)	Training Set (*n* = 140)	Test Set (*n* = 66)	*p*-value ^a^
	AUC (SE)	AUC (SE)	AUC (SE)	
five-miR-panel		0.774 (0.040)	0.735 (0.069)	0.625
	0.745 (0.034)			
six-miR-panel		0.779 (0.040)	0.741 (0.058)	0.587
	0.749 (0.034)			
*p*-value ^b^	0.623	0.930	0.947	

Abbreviations: AUC, area under the receiver operating characteristics curve; SE, standard error of the mean. ^a^ Values indicate that there are no significant differences obtained both with the five-miR-panel and six-miR-panel, using the training or the test set of samples. ^b^ Values indicate that there are no significant differences between the five-miR-panel and six-miR-panel using all samples, the training or test set of samples.

**Table 6 cancers-11-01603-t006:** Construction of a predictive BCR classifier Cox regression in a bootstrapping approach with clinicopathological variables in the 206 patients.

Variable ^a^	Univariate Cox Regression		Multivariate Cox Regression with Significant Variables			
			Full Model ^b^		Reduced Model after Backward Elimination ^c^	
	HR (95% CI)	*p*-value	HR (95% CI)	*p*-value	HR (95% CI)	*p*-value
Age	1.04 (1.00–1.08)	0.029	1.02 (0.98–1.07)	0.231		
PSA	1.04 (1.01–1.07)	0.004	1.02 (0.99–1.05)	0.207		
DRE	1.24 (0.83–1.95)	0.286				
Margin	1.72 (1.16–2.56)	0.008	1.42 (0.64–3.11)	0.548		
pN status	2.66 (0.88–4.10)	0.103	1.14 (0.74–1.75)	0.396		
pT stage	1.12 (1.08–1.17)	<0.0001	1.08 (1.03–1.14)	0.002	1.10 (1.04–1.15)	0.001
ISUP Group	1.37 (1.19–1.57)	<0.0001	1.19 (1.02–1.40)	0.027	1.23 (1.06–1.43)	0.007

^a^ Abbreviations and stratifications of the variables as indicated in Table 1; HR, hazard ratio. ^b^ The full model included all variables of the univariate Cox regression with hazard ratios and *p*-values < 0.200. ^c^ Reduced model after backward elimination with entry *p* < 0.05 and removal *p* > 0.100.

**Table 7 cancers-11-01603-t007:** Improved prediction of biochemical recurrence after radical prostatectomy, using clinicopathological-based tools in combination with the five-miR based panel.

Prediction Tool	Clinicopathological-Based Tool	Clinicopathological-Based Tool Combined with the five-miR-Panel	*p*-value
	AUC (95% CI)	AUC (95% CI)	
*Reference models*			
D’Amico et al. [15]	0.590 (0.519–0.657)	0.759 (0.695–0.816)	<0.0001
CAPRAS	0.692 (0.624–0.754)	0.769 (0.706–0.825)	0.008
NCCN	0.642 (0.572–0.707)	0.757 (0.693–0.814)	0.0005
Stephenson et al. [17]	0.664 (0.595–0.728)	0.747 (0.682–0.805)	0.017
*Present study*			
Full model	0.723 (0.657–0.783)	0.793 (0.731–0.8465)	0.007
Reduced model	0.712 (0.638–0.677)	0.783 (0.720–0.837)	0.007

Abbreviations: AUC, area under the receiver operating characteristics curve; CI, confidence interval; CAPRAS, Cancer of the Prostate Risk Assessment Postsurgical Score, calculated according to [16]; NCCN, National Comprehensive Cancer Network, calculated according to [18]; Full model, according to the Cox regression model described in Table 6, with all clinicopathological factors except for digital rectal examination; Reduced model, according to the Cox regression model described in Table 6 after backward elimination and finally including only the variables pT stage and ISUP Group grade.

**Table 8 cancers-11-01603-t008:** Functional links between miRNAs of the five-miR-panel and prostate cancer.

miRNA [Reference]	Tumor Suppressor Oncogene	Target Gene	Molecular Mechanism
**miR-30c-5p** [79]	Tumor suppressor	ASF/SF2	Inhibition of tumor cell proliferation, promotion of apoptosis through the inhibition of ASF/SF2
[80]	Tumor suppressor	KRAS	Inhibition of tumor cell proliferation, migration, and invasion
[51]	Tumor suppressor	BCL9	Correlation of disease progression via BCL9/Wingles-type signaling
**miR-31-5p** [81]	Tumor suppressor	E2F6	Promotion of apoptosis, reduced prostate cancer growth
[82]	Tumor suppressor	Androgen receptor	Inhibition of cell proliferation and cell cycle
**miR-141-3p** [83]	Tumor suppressor	IGF1R	inhibit malignant transformation of benign prostate epithelial cell via IGF1R-AKT/STAT3 signaling pathway
[84]	Tumor suppressor	TRAF5, TRAF6	Suppression of invasion and migration of PCa cells via inhibiting activation of NF-kB
[85]	Oncogene	KLF9	Regulation of proliferation, spheroid formation, and expression of the stemness factors OCT-4, SOX9, and CCND1
**miR-148a-3p** [86]	Tumor suppressor	RTN4	Inhibition of cell proliferation and cell cycle blocked in G2/M phase	
[87]	Oncogene	CAND1	Increased cell proliferation	
[88]	Tumor suppressor	MSK1	Inhibition of cell growth, migration, and invasion	
**miR-221-3p** [89]	Tumor suppressor	Runx1, Runx2	Inhibition of prostate tumorigenesis
[82]	Tumor suppressor	Androgen receptor and its receptor coactivators	Inhibition of cell proliferation and targeted key oncogenic pathways, including cell cycle
[90]	Oncogene	PI3K/AKT, P53	Associated with signal transduction and cell communication

Abbreviations: ASF/SF2, serine and arginine rich splicing factor 1; KRAS, KRAS proto-oncogene, GTPase; BCL9, BCL9 transcription coactivator; E2F6, E2F transcription factor 6; IGF1R, insulin like growth factor 1 receptor; AKT/STAT3, serine-threonine kinase/ signal transducer and activator of transcription; TRAF5, TNF receptor associated factor 5; TRAF6, TNF receptor associated factor 6; NF-kB, nuclear factor kappa B subunit 1; KLF9, Kruppel like factor 9; OCT-4, organic cation/carnitine transporter4; SOX9, SRY-box transcription factor 9; CCND1, cyclin D1; RTN4, reticulon 4; CAND1, cullin associated and neddylation dissociated 1; MSK1, mitogen- and stress-activated kinase 1; Runx1, runt related transcription factor 1; Runx2, runt related transcription factor 1; P13K/AKT, phosphatidylinositol 3-kinase/serine-threonine kinase; P53, tumor protein 53.

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
