# Peer review of "A Novel Predictor Tool of Biochemical Recurrence after Radical Prostatectomy Based on a Five-MicroRNA Tissue Signature"

_cancers, 2019, doi:10.3390/cancers11101603_

Round 1

Reviewer 1 Report

The authors adequately addressed my comments.

Reviewer 2 Report

Thank you. The authors have addressed my concern and I am happy for the paper to be published.

This manuscript is a resubmission of an earlier submission. The following is a list of the peer review reports and author responses from that submission.

Round 1

Reviewer 1 Report

Well thought out retrospective review. Would have been stronger if external validation but in saying this does add a novel method to analysis of risk in patients with prostate cancer. Science and statistical analysis appear sound. Micro-RNA approach is interesting.

Study gives interesting data into this area. Hopefully in future can be confirmed in a multi-institutional format in a prospective fashion with external validation of methods.

Reviewer 2 Report

This manuscript describes an evaluation of 20 miRNAs with respect to biochemical recurrence in prostate cancer. I found it to be a compelling story and the prediction accuracy is impressive. That said, the manuscript would benefit from addressing the following:

General:

The manuscript requires some minor editing for grammar.

Introduction

Regarding the incidence of prostate cancer worldwide, why not cite more recent data than those from 2012? Similarly, some of the references regarding the frequency of BCR following radical prostatectomy strike me as rather outdated. Isn’t it possible that advances in surgical procedures since the early 2000s have altered the prevalence of BCR? I hate to feel so repetitive, but the evidence that clinicians consider increasing PSA levels to be a first sign of progression is 20 years old! If nothing has changed since then, perhaps the authors could explicitly state as such. I worry that the authors haven’t sufficiently made the case for BCR as an interesting endpoint. The prevalences of clinically manifested recurrence and PCa-specific mortality in individuals who experience BCR aren’t particularly informative without information about comparable prevalences in individuals who do not experience BCR. I doubt that the Meurs meta-analysis indicated that the CAPRAS had zero predictive ability with respect to five-year recurrence-free survival in PCa patients.

Results

The flow chart does not indicate whether the adjacent normal tissue was sourced from individuals who did or did not experience BCR. Can the authors provide some justification for grouping pN0 and pNx together? Table 1 indicates that the authors used the Mann-Whitney U test, Chi-square, or Fisher's exact test to evaluate differences between the two groups. I wonder whether they might consider a test that accounts for trend (e.g., Cochran-Armitage) when evaluating ordinal variables? The authors should specify the number of miRNAs from among which the 20 included were selected. I.e., how many miRNAs were evaluated for possible inclusion? I don’t know that Figure 2 requires panels for the miRNAs that were not differentially expressed. A significance threshold of 0.05 is problematic given the number of tests that the investigators ran. How many associations would be significant were the authors to account for multiple testing? In Section 2.3, the authors should explicitly describe the number of significant correlations between clinicopathological factors and the particular miRNAs that were differentially associated with BCR status. From which analyses were the p-values <0.2 for the miRNAs selected for the Cox models? Adjacent vs. normal or BCR vs. no BCR. It’s easier for the reader not to have to refer back in the article. To what extent did the 6-miR panel overlap with the 8 miRNAs that showed differential associations according to BCR status? What are the p-values presented in Table 5?

Discussion

The third paragraph of the Discussion does not provide interpretation beyond what is presented in the Results. For findings that did not confirm those from previous studies (e.g., an association of miR-30c-5p levels with Gleason and stage, different miRNAs relative to Nam, et al. and Kristensen, et al.), can the authors comment on why they might have seen differences? How do the authors expect that their panel would compare to the other tissue-based tests outlined in the Discussion? When describing potential mechanisms that could mediate miRNA associations with PCa outcomes, can the authors comment on directionality? I.e., Do the directions of the associations in the study make sense given the possible mechanisms? Can the authors comment on any potential inherent biases in the study design? Could it have any impact that patients were required to remain under evaluation until October 2018? And are there any disadvantages of implementing Cox models when the authors selected on outcomes? What would it look like to evaluate miRNAs as part of clinical practice? And to what extent would doing so improve PCa outcomes? Perhaps questions that the authors could suggest as future directions.

Materials and Methods

Samples were reviewed by two expert uropathologists to what end?

Reviewer 3 Report

This manuscript by Zhao et al describes a 5 micro-RNA signature to predict biochemical recurrence in prostate cancer. The articles is well written, the studies well done and is of interest to the field. 

To improve I would suggest adding a table summarising what is already known about the 5 micro-RNAs (function and link to cancer/prostate cancer if known). This would help highlight this key information to the reader.
